# Valorisation of Grape Stems as a Source of Phenolic Antioxidants by Using a Sustainable Extraction Methodology

**DOI:** 10.3390/foods9050604

**Published:** 2020-05-08

**Authors:** Juan Antonio Nieto, Susana Santoyo, Marin Prodanov, Guillermo Reglero, Laura Jaime

**Affiliations:** 1Institute of Food Science Research (CIAL), Universidad Autónoma de Madrid (CEI, UAM-CSIC), 28049 Madrid, Spain; juan.nieto@uam.es (J.A.N.); susana.santoyo@uam.es (S.S.); marin.prodanov@uam.es (M.P.); guillermo.reglero@uam.es (G.R.); 2IMDEA-Food Institute, Campus de Cantoblanco, 28049 Madrid, Spain

**Keywords:** grape stem, phenolic compounds, central composite rotatable design, antioxidant activity, sustainable food systems, pressurized liquid extraction, side streams valorisation

## Abstract

Pressurized liquid extraction with ethanol:water mixtures was proposed for obtaining phenolic antioxidants from grape stems. The optimal extraction conditions were elucidated by using a central composite rotatable design (solvent (X_1_, 0–100% ethanol:water *v*/*v*), temperature (X_2_, 40–120 °C) and time (X_3_, 1–11 min)). Response surface methodology determined 30% ethanol:water, 120 °C and 10 min as the optimal extraction conditions regarding total phenolic content (TPC) (185.3 ± 2.9 mg gallic acid/g of extract) and antioxidant activity (3.55 ± 0.21 mmol Trolox/g, 1.22 ± 0.06 mmol Trolox/g and 1.48 ± 0.17 mmol Trolox/g of extract in ABTS, DPPH and ORAC methodologies, respectively). The antioxidant activity was attributed to total polymer procyanidins and flavan-3-ol monomers and oligomers, although other phenolic compound contributions should not be ruled out. Forty-two phenolic compounds were identified in the optimal extract, mainly polymer procyanidins and, to a lesser extent, monomers and oligomers of flavan-3-ols, quercetin-3-*O*-glucuronide, ε-viniferin, gallic and caftaric acid. Ethyl gallate, ellagic acid, protocatechuic aldehyde, delphinidin-7-*O*-glucoside and cyanidin-3-*O*-glucoside were reported for the first time in grape stem extracts. In conclusion, this study highlights the use of this winery side stream as a source of antioxidants within a sustainable food system.

## 1. Introduction

Nowadays, there is increasing concern about the sustainability of food production, including an environmentally friendly use of food by-products. In this context, one of the main goals of FAO is the promotion of sustainable food systems [1]. Accordingly, the exploitation of vegetable by-products as a source of bioactive compounds represents a promising opportunity to obtain added-value products for food or pharma industries. Particularly, wine production generates thousands of tons of solid organic waste, where grape stems represent up to 5% (*w*/*w*) of the processed grapes, being approximately 25% of the total by-products generated by the wine industry [2].

Whereas several studies have been focused on *Vitis vinifera* L. grape skins and seeds, less attention has been paid to grape stems as source of useful bioproducts [3,4]. Research papers focused on stem phenolic composition pointed out the presence of a high content of flavan-3-ol monomers (catechin and epicatechin) and, to a lesser extent, stilbenes (mainly resveratrol and ε-viniferin), as well as different phenolic acids and flavonols [5,6,7]. These compounds are known for their important biological activities; therefore, the use of stem by-products as a source of bioactive compounds has become a great opportunity to obtain functional ingredients in a sustainable way [4,8]. Nevertheless, there is a lack of studies with a comprehensive description of the phenolic composition of grape stem extracts. Thus, some studies are only focused on stilbene compounds [9,10], flavan-3-ol composition [11], their main phenolic constituents [3,6], or other specific groups [12,13]. However, there is scarce information about the proanthocyanidin fraction [12,14].

Extraction procedure is an important step to consider in recovery of bioactive compounds from plant sources. As conventional solid–liquid extraction (SLE) procedures usually have different drawbacks (such as long extraction time, intensive labour procedures, large volumes of solvents needed, and low extraction yields), many alternative techniques have been emerged in the last two decades [15]. In this regard, pressurized liquid extraction (PLE), ultrasound assisted extraction (UAE), microwave assisted extraction (MAE) and supercritical fluid extraction (SFE) have been proposed for extraction of bioactive compounds. Specifically, PLE has been used for extraction of phenolic compounds from a wide range of plant materials, e.g., hops, aromatic plants, grapes, as well as microalgae. Methanol, water, ethanol, acetone or their aqueous mixtures have been commonly used up to 170 °C in a light- and oxygen-free environment, with a greater efficiency than SLE procedures [16,17]. Development of an extraction methodology implies the establishment of optimal extraction conditions for a specific compound or group of compounds. For this purpose, the use of experimental designs, along with response surface methodology (RSM), has been proposed as a useful tool that enables the analysis of the influence of different extraction factors and allows reducing the number of experimental trials. Accordingly, it has been successfully used for improving phenolic extraction procedures from vegetable materials [12,13].

Regarding grape stem by-products, SLE with organic solvents has been commonly used for extraction of phenolic compounds [18]. However, alternative extraction techniques such as UAE [19], SFE [20], MAE [21] and PLE [22] have been scarcely studied. Moreover, only a few studies have focused on the use of green solvents, such as pure sub- and supercritical ethanol [23] or aqueous-ethanol mixtures with conventional SLE [12,13].

Therefore, to the best of our knowledge, a green extraction procedure based on the application of PLE to extraction of phenolic antioxidants from grape stems has not been established yet. Moreover, the relationship between phenolic composition and antioxidant activity should be established.

Hence, the aim of the present study was to develop a sustainable extraction procedure for valorisation of grape stems as a source of phenolic antioxidant compounds by using PLE and ethanol:water mixtures as green solvents. For this purpose, an experimental design along with RSM was used to optimize the extraction procedure (extraction solvent, temperature and time). Additionally, an extensive analysis of the phenolic composition of the extracts was done.

## 2. Materials and Methods

### 2.1. Chemicals and Reagents

Acetonitrile and formic acid, HPLC quality, were supplied by Labscan (Dublin, Ireland) and Acros Organic (Belgium), respectively. Protocatechuic acid, vanillic acid, syringic acid, caffeic acid, p-coumaric acid, 3-coumaric acid, ethyl gallate, 3,5,4’-trihydroxystilbene-3-*O*-β-d-glucoside (*trans*-piceid), (+)-catechin, (−)-epicatechin, epicatechin gallate, procyanidin B_1_, procyanidin B_2_, procyanidin B_3_, quercetin-3-*O*-galactoside, quercetin-3-*O*-rutinoside, quercetin-3-*O*-glucuronide, quercetin-3-*O*-glucoside, quercetin dihydrate, delphinidin-3-*O*-glucoside, cyanidin-3-*O*-glucoside and malvidin-3-*O*-glucoside were purchased from Extrasynthèse (Genay, France). Gallic acid, 4-hydroxybenzoic acid, *trans*-caftaric acid, *trans*-ferulic acid, ellagic acid, protocatechuic aldehyde, *trans*-resveratrol, 6-hydroxy-2,5,7,8-tetramethylchromane-2-carboxylic acid (Trolox), potassium persulfate, 2,2’-azinobis(3-ethylbenzothiazoline-6-sulphonic acid) diammonium salt (ABTS), 2,2-diphenyl-1-picrylhydrazyl (DPPH), 1 M phosphate buffer, fluorescein sodium and 2,2’-azobis(2-methylpropionamidine) dihydrochloride (AAPH) were obtained from Sigma-Aldrich (Madrid, Spain). Disodium carbonate, Folin–Ciocalteu reagent, methanol and ethanol were from Panreac (Barcelona, Spain).

### 2.2. Plant Material

Grape stems (*Vitis vinifera* L. cv. Merlot) were provided by Instituto Madrileño de Investigación y Desarrollo Rural, Agrario y Alimentario (IMIDRA, Spain). Stems were separated manually. Raw fresh stems were dried at 40 °C for 48 h in an air bath Stuart S150 (Stuart, UK). Thereafter, dried material was ground in a blender and the resulting powder was sieved to a ≤ 1 mm particle size and stored in a closed bag at −20 °C until further use.

### 2.3. Experimental Design

A central composite rotatable design (CCRD) was used to evaluate the influence of three independent variables (factors), i.e., solvent composition (X_1_, 0%–100% ethanol), extraction temperature (X_2_, 40–120 °C) and extraction time (X_3_, 1–11 min) on the yield, antioxidant activity (ABTS, DPPH), total phenolic content (TPC) and flavan-3-ols content (mono-oligomers and polymers), measured by means of NP-HPLC, after PLE extraction. Therefore, a total of 19 experimental conditions were established at five levels (2^3^ points of the full factorial design + 6 star points + 5 central points) (Table 1). To determine the influence of the different experimental factors, response surface methodology (RSM) was used, and regression analysis of the response variable data was performed and fitted to a quadratic polynomial model as shown in the following equation:*Y* = *β*_0_ + *β*_1_*Et* + *β*_2_*T* + *β*_3_*t* + *β*_1,1_*Et*^2^ + **β**_2,2_*T*^2^ + *β*_3,3_*t*^2^ + *β*_1,2_*EtT* + *β*_1,3_*Ett* + *β*_2,3_*Tt* + *ε*,(1)
where *Y* is the response variable; *β*_0_ the independent term; *β*_1_, *β*_2_ and *β*_3_ the linear coefficients; *β*_1,1_, *β*_2,2_ and *β*_3,3_ the quadratic coefficients; *β*_1,2_, *β*_1,3_ and *β*_2,3_ the interaction factor coefficients and *ε* the experimental error. The goodness of fit was evaluated according to the determination coefficient (R^2^), the residual standard deviation (RSD), and the lack of fit test provided by the analysis of variance (ANOVA).

### 2.4. PLE

Extractions were performed by using an ASE 350 extractor from Dionex Corporation (Sunnyvale, CA, USA). Dry stem powder (1 g) was mixed in a ceramic mortar with 1 g of diatomaceous earth (Dionex, Sunnyvale, CA, USA). The solid mixture was loaded in an 11-mL extraction cell. Two cellulose filters (Dionex, Sunnyvale, CA, USA) were placed at the bottom of the cell to prevent the clogging of the system. The cell was automatically filled with the proper solvent to a pressure of 1500 psi. The extraction cell heat-up time was fitted according to the applied extraction temperature (e.g., 5 min, when the extraction temperature was 40 °C). Subsequently, a static extraction was performed. Afterwards, the cell was rinsed (60% cell volume) and solvent was purged from the cell with pressurized N_2_ gas for 90 s. A rinse of the complete system was made between extractions. Ethanol from all extracts was removed by vacuum evaporation at 37 °C in an IKA RV 10 control (IKA, Staufen, Germany), followed by freeze-drying (Telstar Lyobeta 15 equipment; Telstar, Madrid, Spain). Powder extracts were kept at −20 °C until analysis.

### 2.5. Identification and Quantification of Phenolic Compounds by RP-HPLC-PAD-MS

Individual phenolics were determined by reversed-phase high-performance liquid chromatography (RP-HPLC) as described by Grases et al. [24], adapted for grape stems. Chromatographic analyses were carried out in a C18 ACE RP18-AR (150 mm × 4.6 mm, 3 µm particle size) (Symta, Madrid, Spain) protected by a guard column ACE 3 C18-AR (7 mm × 13 mm) packed with the same stationary phase. The column oven was set at 30 °C and sample injection volume was 40 µL. A gradient consisting of 1% aqueous formic acid (solvent A) and acetonitrile containing 1% formic acid (solvent B) was used at a flow rate of 0.6 mL/min. The elution programme was applied as follows: 0 min, 0% B; 80 min, 20% B; 115 min, 28% B; 125 min, 50% B; 135 min, 100% B; 145 min, 100% B; and finally 10 min to recover initial chromatographic conditions. Chromatographic separation was carried out in an Agilent HPLC 1260 series equipped with a photodiode array detector (PAD) coupled online to an ion trap mass selective detector with an electrospray ionization source (Agilent Technologies Inc., Santa Clara, CA, USA). The system was controlled by ChemStation software (Agilent, vers. 6.8). Chromatograms were recorded at 280, 320, 360 and 520 nm. The electrospray ionization (ESI) parameters were as follows: drying gas (N_2_) flow and temperature, at 11 L min^−1^ and 350 °C, respectively; nebulizer pressure 65.0 psi; and capillary voltage 4 kV. The full-scan mass covered the range from *m*/*z* 100 to 2000 uma. Mass spectrometry data were acquired in positive ionisation mode for anthocyanins and negative ionisation mode for the other phenolic compounds. Prior to injection, all samples were diluted in 1 mL of ultrapure water:methanol (1:1) mixture and then filtered by a 0.45 µm PVDF membrane filter. 

Phenolic compounds were identified according to their retention time, mass-to-charge ratio (*m*/*z*) of their molecular ions and UV/Vis spectrum by chromatographic comparison with analytical reference substances. Procyanidin dimers B_4_ and B_7_ were purified from natural extracts by high-speed countercurrent chromatography (HSCCC) as described in Grases et al. [24]. Procyanidin trimer C_1_ was identified by the use of a purified OPC-rich extract from cocoa as a complex reference substance, according to Prodanov et al. [25]. The rest of phenolic compounds were tentatively identified on base of their peak retention time, *m*/*z* of their molecular ions and diagnostic fragments, UV/Vis spectrum and/or data from Scifinder database and literature.

For quantification purposes, PAD was used. Chromatographic conditions were similar to those already described for identification purposes. Phenolic compounds were quantified by calibration curves of their respective reference solutions, except for procyanidin oligomers, apart from B_2_, and resveratrol derivatives that were quantified from procyanidin B_1_ and resveratrol calibration curves, respectively. Hydroxybenzoic acids and flavan-3-ols were quantified at 280 nm, hydroxycinnamic acids and stilbenes at 320 nm, flavonols at 360 nm and anthocyanins at 520 nm. Samples were analysed at least in triplicate.

### 2.6. Analysis of Total Flavan-3-ol Monomers and Oligomers and Total Polymers by NP-HPLC

Analyses of total flavan-3-ol monomers and oligomers and total polymers, based on increasing order of their molecular masses (degree of polymerization), were performed by NP-HPLC following Muñoz-Labrador et al. [26] elution programme. A Kromasil 60 DIOL column (250 mm × 4.6 mm, 5 µm particle size; AzkoNobel, Amsterdam, Netherland) was used with a security guard Lichrospher Diol-5 (7 mm × 13 mm) cartridge packed with the same material. Column temperature was kept at 35 °C. Solvent (A) was 2% acetic acid in acetonitrile, solvent (B) was methanol containing 2% acetic acid and 3% water, and solvent C was a 2% aqueous acetic acid. A constant flow rate of 0.8 mL min^−1^ was used. Volume sample injection was 10 µL. Chromatographic analyses were performed using an Agilent Infinity 1260 liquid chromatograph system.

Flanvan-3-ol monomer and oligomers were identified according to their retention time and UV/Vis spectrum in comparison with a purified flavan-3-ol oligomer from cocoa-rich extract, used here as a complex reference substance for oligomer procyanidins of up to octamers. Total polymer procinanidins eluted as a singular peak at the end of the chromatogram [26]. Both mono- and oligomers and total polymers were quantified by means of catechin calibration curve at 280 nm. The results were expressed as mg of catechin equivalent (CE)/g extract. All analyses were done in triplicate.

### 2.7. Determination of Mean Degree of Procyanidin Polymerization (mDP)

Flavan-3-ol procyanidins were isolated in a minicolumn assembly-line system (minicolumn cartridge C18 Sep-Pack and tC18 Sep-Pack from Waters, Milford, MA, USA) as previously described by Sun et al. [27]. After that, degradation of isolated procyanidins was done by acid-catalysed degradation using toluene-α-tiol. Quantification of degradation products and mDP were conducted by RP-HPLC-PAD. All analyses were done in triplicate.

### 2.8. Total Phenolic Content (TPC)

Total phenolic content was determined according to the Folin–Ciocalteu reagent method [28]. The results were expressed as mg of gallic acid equivalents (GAE)/g extract. All analyses were done in triplicate.

### 2.9. Antioxidant Activity

ABTS^+^ and DPPH radical scavenging assays were carried out according to the original method described by Re et al. [29] and Brand-Williams et al. [30], respectively, and both results were expressed as TEAC value (mmol Trolox/g extract). 

ORAC assay was carried out using the method of Huang et al. [31] with some modifications. Briefly, 150 µL of fluorescein solution (8 × 10^−8^ M fluorescein in 0.075 M phosphate buffer) was mixed with 25 µL of sample, phosphate buffer (blank) or Trolox solution (100, 80, 60, 40, 20 and 10 µmolar solution), or 50 µL of phosphate buffer (control) in a 96 wells plate. The reaction was carried out by adding 25 µL of a fresh AAPH solution (165.94 mmol AAPH in phosphate buffer) at 37 °C, except for control wells. The mixture was shaken for 8 s and fluorescence intensity was monitored for 120 min (485 nm and 520 nm for excitation and emission wavelength, respectively). ORAC values were expressed as mmol Trolox/g extract.

Antioxidant analyses were done in triplicate.

### 2.10. Statistical Analyses

The statistical analysis of CCR experimental design data was carried out by RSM with the statistical program Statgraphics Centurion XVI (Statistical Graphics Corp., Warrenton, VA, USA). Correlation coefficients between the different experimental data were performed using Pearson’s test (*p* ≤ 0.05). Moreover, a principal component analysis (PCA) was conducted for correlation between response variables.

## 3. Results and Discussion

### 3.1. Experimental Model Fitting

The present study was conducted to evaluate the optimal conditions for the extraction of phenolic antioxidants from Merlot’s grape stems (*Vitis vinifera* L.). This variety is one of the most representative grape varieties as it is one of the most widespread cultivars. For this purpose, a central composite rotatable design was applied as displayed in Table 2. Three independent variables or factors, namely, ethanol concentration (X_1_, 0–100%), temperature (X_2_, 40 °C–120 °C) and time (X_3_, 1–11 min), were studied to assess their influence on antioxidant activity, flavan-3-ol monomers and oligomers, polymer procyanidins and total phenolic compounds. These factors are considered as the main extraction-independent variables, at the expense of others less significant, such as solid–solvent ratio or solvent pH [13,32]. Aqueous methanol has been proposed as the most suitable solvent for the extraction of phenolic compounds from grape products [11,22]. Nevertheless, this toxic solvent is currently being replace by aqueous ethanol [6,12]. Its high efficiency and GRAS status makes it suitable for food or pharmaceutical applications [12]. Heating up to 200 °C has been commonly used in PLE extractions, since higher temperatures lead to a greater extraction yield [23]. Nevertheless, values above 120 °C should be avoided as they may cause degradation of phenolic compounds [16]. Accordingly, the maximal temperature value was set at 120 °C in the present study. Moreover, the extraction time variable was limited in low values, since higher extraction time may promote phenolic degradation without an enhancement of extraction yields [17,32]. Table 2 shows response variable data corresponding to each experimental condition.

The regression coefficients of linear, quadratic and interaction terms of the experimental factors were calculated by fixing the experimental values of response variables to a quadratic linear regression model. The effect of each term in the model and its statistical significance on the response variables were analysed from the standardized Pareto chart (data not shown). The quadratic and interaction terms not significantly different from zero (*p* ≤ 0.05) were excluded from the model, and the mathematical model was refitted by multiple linear regression (MLR), resulting in the polynomial equations shown in Table 3.

These equations suggest that RSM was successfully applied for the optimisation of the considered variables. The models did not show significant lack of fit (*p* > 0.05), indicating well-fitting models for yield, TPC, ABTS and DPPH, opposite to flavan-3-ol monomers and oligomers, and polymer procyanidin behaviour. However, determination coefficients (R^2^) for all the studied variables were over 0.90. Therefore, the proposed models could be used as an approach to the real behaviour of these compounds regarding these extraction parameters. Concerning extraction yields, the obtained data exhibited a very good fit for a quadratic model. Moreover, all the experimental factors were significant, mainly ethanol proportion in a quadratic manner (Table 3). In addition, some significant interaction factors were found. Response surface plot showed that an increase of ethanol proportion caused an extraction yield rise of up to almost 25% (Figure 1). Therefore, higher and lower ethanol:water ratios led to a decrease in the extraction yield, more markedly at high ethanol proportions (Figure 1A,B). Moreover, temperature caused a linear effect, that is, extraction yield was enhanced linearly when temperature increased (Figure 1A). On the other hand, a weak quadratic influence of extraction time was noticed.

Similar results were found regarding the total phenolic content (TPC) and antioxidant activity (ABTS, DPPH), where good fitting models for these response variables were established (Table 3). Furthermore, extraction solvent was the most important factor, showing a quadratic effect, together with a linear effect of temperature (Figure 2A, Figure 3A and Figure 4A). Although time and some interaction factors resulted in meaningful effects, a lower contribution to response variables was determined for these parameters (Figure 2B, Figure 3B and Figure 4B), where extraction time significance is generally linked to temperature [32]. The optimum ethanol concentration effect was determined to be close to 30%; meanwhile, 120 °C allowed reaching the highest response variable values. Similar results were found in other studies, where ethanol:water mixtures’ behaviour was analysed [12,13]. These authors indicated that extraction solvent was the main factor for both antioxidant activity and TPC of the extracts, showing a quadratic main effect. The optimum ethanol concentration in ethanol:water mixtures to achieve the maximum phenolic extraction, and therefore the greatest antioxidant capacity, is generally observed between 30% and 80% [4,12,13]. In accordance with the present study, the optimization of conventional solid–liquid extraction for two grape stem samples through RSM determined that ethanol and temperature were the main factors during phenolic extractions. The optimum ethanol:water concentration was determined as 57.9% and 63.8%. Although a strong negative and quadratic effect was also observed in these studies for ethanol concentration effect, noticeably higher amounts of ethanol were found [12]. The lower optimum ethanol content in the PLE extraction solvent of the present study (30%) is probably due to the improvement extraction capacities of the solvents because of the pressure and higher temperatures applied during the extraction process. Moreover, PLE modified the extraction capacities of solvents by reducing their polarity [17] and enhancing the extraction of low polar compounds, such as phenolic compounds. Therefore, PLE reduces the required ethanol:water proportion compared to conventional solid–liquid extraction [12,13] or ultrasound extraction [4]. Besides, temperature increase is generally associated with enhancements of phenolic compound extraction [12,13]. Since PLE allows increasing the temperature over the solvent boiling point, the extraction capacity of the used solvents is generally enhanced [17]. Because of that, optimum extraction time is reduced when PLE is used in comparison with conventional extraction [12,13].

Furthermore, temperature was found to be an important but controversial factor, regarding phenolic compound extraction and antioxidant activity. In general, a high extraction temperature is correlated with an increase in the solubility of phenolic compounds from the matrix [33], as it reduces solvent viscosity and enhances solvent penetration [34]. On the other hand, high temperatures may lead to breakdown of thermolabile compounds [13]. Nevertheless, this shortcoming could be avoided by using high extraction temperatures (100–120 °C) at a short extraction time [35]. Accordingly, in the present study, a short extraction time (up to 10 min) at 120 °C avoided thermal degradation of some phenolic compounds.

Regarding total flavan-3-ol monomers and oligomers or total polymers of procyanidin content, experimental data did not fit with the quadratic proposed model, suggesting more complex behaviour. Nevertheless, according to response surface plots, extraction solvent resulted as the main factor in the total mono-oligomer demeanour, whereas temperature and time were remarkably less meaningful. Ethanol seems to enhance flavanol-3-ol monomer and oligomer extraction (Figure 5A), while temperature or time influences are less clear (Figure 5A,B). Opposite to that, higher ethanol proportions seem to reduce the extraction of total polymer procyanidins, while temperature increased their extraction yield (Figure 6A). In this case, although extraction time turned out to show a positive tendency, an important interaction between time and the rest of studied factors distorted the effect of time into an overall negative trend (Figure 6A,B). Moreover, temperature increase resulted in a higher extraction of polymers, while ethanol had a suppressive effect, but enhanced the flavan-3-ol monomer and oligomer yield. Sun and Spranger [36] indicated that temperature allowed higher proanthocyanidin extraction rates if the temperature did not reach a degradation point. Likewise, solvent mixtures with higher polarities improve the extraction of these compounds, disrupting the bonds between phenolic compounds and the matrix. In this sense, Karvela et al. [13] found that greater flavanol monomers contents were reached at 60% of ethanol:water, whereas oligomers and polymer proanthocyanidins were achieved at 44% and 55%, respectively.

Nevertheless, it is important to highlight that, in all cases, individual optimisation of extraction conditions is required as grape variety, agro- and weather conditions or plant material are other factors that are not taken into consideration and can have significant influence on phenol recovery [12].

### 3.2. Optimal Conditions and Validation of the Developed Model

Optimal experimental conditions were achieved for extraction yield, TPC and antioxidant activity since these response variables were fitted to the proposed model. As can be seen in Table 4, only slight differences were found regarding the optimal conditions of these variables. A tight range of ethanol concentrations (22%–30%) was found as the optimum, while the highest temperature (120 °C) was optimal in every response variable. Moreover, high extraction time showed better results, except for extraction yield and TPC, where a slight decrease was shown at 11 min. Therefore, 30% ethanol, 120 °C and 10 min were selected as the most suitable extraction conditions in order to obtain an extract with the highest contents of phenolic antioxidants (optimum extract) by PLE from Merlot grape stem.

Under such optimal conditions, the statistical model predicted an extraction yield of 29.2%, 192.4 mg GAE/g extract (TPC), and TEAC values of 3.81 and 1.31 mmol Trolox/g extrac regarding the ABTS and DPPH methods, respectively. To corroborate these values, additional extractions were made at the optimal extraction conditions (Table 4). The results display that optimum extract showed values very close to the predicted ones, validating the proposed model. In addition, ORAC assay was carried out in order to perform a deeper antioxidant characterization of this optimum extract, with an ORAC value of 1.48 ± 0.17 mmol Trolox/g extract. Moreover, 26.8 ± 0.4 mg of mono-oligomers and 79.17 ± 1.36 mg of proanthocyanidin polymers were quantified in this extract.

These optimal experimental conditions allowed obtaining an extraction yield, TPC and antioxidant activity in agreement with the wide range of values determined by González-Centeno et al. [22] for several grape stem varieties of *Vitis vinifera*, or even slightly higher results regarding PLE Merlot extract. However, it should be noted that these authors proposed substantially different extraction conditions.

Experimental designs have been proposed previously to find optimal extraction conditions for grape stems [12,13]. In these studies, ethanol:water mixtures, time, temperature or pH were studied as experimental factor using SLE. Slightly higher TPC, along with greater TEAC values, was found in the present study. It is worth mentioning that a lower ethanol:water proportion was required at optimal extraction conditions compared to SLE [12,13]. As has been mentioned before, even if the grape variety or environmental conditions should affect this behaviour, this result could be ascribed to an electric constant decrease in the extraction solvent at high pressure [37]. Furthermore, shorter extraction time and less consumption of solvents were used for PLE than for SLE.

Moreover, similar or lower values of TPC and antioxidant activity were observed when data presented in this study were compared with other SLE or PLE extracts from grape stems [8,18,22]. However, regarding these parameters, the few data in the literature concerning grape stem extracts show a wide range of values. In this regard, differences caused by ripening stage, grape variety, geographic factors, climatological factors or oenological practices, as well as different extraction procedures applied, should be considered [38,39].

### 3.3. Correlation between Response Variables

Correlations between TPC and TEAC values (ABTS and DPPH) were established in order to confirm the influence of phenolic compounds on the antioxidant activity of the extracts. Antioxidant activity, measured by two methods in the present study (ABTS and DPPH) resulted in a strong correlation with TPC, achieving r = 0.993 and r = 0.987, respectively (*p* ≤ 0.001). Besides, antioxidant activities obtained by both methods showed a high correlation between them (r = 0.993; *p* ≤ 0.001)). This confirms that phenolic compounds are the main factor responsible for the extract antioxidant activity, according to the similar optimal conditions predicted by RSM models. In addition, Pearson’s test indicated a good correlation between TPC and the two fractions of phenolic compounds, flavan-3-ol monomers and oligomers (r = 0.557; *p* ≤ 0.05) and polymers (r = 0.586; *p* ≤ 0.01). Furthermore, statistical correlations between antioxidant activity and flavan-3-ol monomers and oligomers were found (r = 0.571; *p* ≤ 0.05 for ABTS method, and r = 0.617; *p* ≤ 0.01 for DPPH method). However, correlations turned out to be stronger with polymer procyanidins, being r = 0.621 (*p* ≤ 0.01) for ABTS and r = 0.637 (*p* ≤ 0.05) for the DPPH method. These results reveal that the antioxidant activity of the grape stem extracts could be ascribed to their general phenolic content [38], although particular contributions of phenolic groups, such as polymer proanthocyanidins, could be higher [13].

Additionally, a PCA was carried out to understand correlations between procyanidins and antioxidant activity. The principal component of the analysis explains a 90.79% of the samples. The antioxidant activity, determined using the ABTS method, was mainly explained by PC1 (76.48%) whereas PC2 contributed to a lesser extent (14.32%) (Figure 7A). Therefore, PCA analyses showed a strong correlation between the polymer content and the total antioxidant activity of the samples. Therefore, those samples characterised by higher amounts of polymers, being samples 7, 12 and the optimum extract, showed greater antioxidant activity. These samples were characterised by the use of high temperatures during PLE extraction (104 °C, 120 °C and 120 °C, respectively), being in concordance with the polynomial equations of the fitted models (Table 3). Additionally, the proximity of samples 3 and 14 in the PCA graph, as well as sample 7, evidences the influence of the extraction time. Similar results were observed when PCA analysis was conducted for DPPH values (Figure 7B), explaining 89.65% of the samples.

### 3.4. Phenolic Composition of the Optimum Extract

Forty-two phenolic compounds were identified by HPLC-PAD-MS. Optimal PLE grape stem extract showed a complex composition of phenolic compounds, including phenolic acids, stilbenes, flavonols and, especially, flavanols (Table 5). Regarding phenolic acids, gallic and caftaric acids were the main hydroxybenzoic and hydroxycinnamic acids of the extract, followed by vanillic and syringic acid [19]. In addition, stilbenes were identified, including *trans*-resveratrol, *ε*-viniferin, *trans*-resveratrol-glucoside (piceid), along with different dimmers and trimers of *trans*- and *cis*-resveratrol [9,10,40].

Numerous flavan-3-ols were also determined, including monomers, dimers and oligomers [40]. Catechin was the main monomeric compound, followed by epicatechin, whereas dimer B_1_ turned out to be the highest dimer compound [11].

Moreover, different flavonols were quantified, mainly as quercetin derivatives. The most remarkable compound corresponded to quercetin-3-*O*-glucuronide, followed by quercetin-3-*O*-glucoside [40]. These forms of quercetin are the main flavonols of grape stems, along with others such as quercetin-3-*O*-rutinoside and quercetin-3-*O*-galactoside [8,19,40]. In addition, low quantities of different anthocyanins where detected in the extract, malvidin-3-*O*-glucoside being the most abundant of this group [8].

It is remarkable that, as far as we know, it is the first time that compounds such as ethyl gallate, ellagic acid, delphinidin-7-*O*-glucoside or cyanidin-3-*O*-glucoside have been identified in grape stem extracts, although ethyl gallate has been previously reported in grape seed extracts [25].

Just a few articles depict the composition of proanthocyanidin fraction in grape stem extracts. In the present study, 80 mg of CE/g extract was quantified as procyanidins according to the procedure of Sun et al. [27]. Procyanidin characterization revealed a mDP of 12 units. Structural composition showed catechin as a predominant terminal unit and epicatechin as a principal extension unit, where approximately 12% of the total constituent units of procyanidins were galloylated (Table 6). Focusing on general composition, epicatechin was the main monomer unit (72.3%), followed by catechin (14.6%), epicatechin gallate (12.4%) and, to a lesser extent, epigallocatechin (0.7%). All these results are consistent with the stem proanthocyanidin fraction described in literature [14,22], and particularly with Merlot’s proanthocyanidins [7]. Nevertheless, slightly higher mDP was found in the present study. This result might be attributable to the effect of pressure extraction conditions used in this study, allowing a higher penetration of the solvent during extraction process.

## 4. Conclusions

Ethanol:water mixture (30%), 120 °C and 10 min turn out to be the optimal extraction conditions of the environmentally friendly PLE process carried out in this study. These conditions lead to obtaining an extract with a high phenolic content and a remarkable antioxidant activity from Merlot grape stems. This methodology allows reducing solvent volume, ethanol concentration and extraction time compared to conventional solid–liquid extraction. This study shows that this by-product is an interesting and undervalued source of procyanidins, and to a lesser extent, stilbene and flavonol derivatives. Moreover, although the contribution of polymer procyanidins to the antioxidant activity of the extract is established, the role played by other phenolics should not be ruled out. Therefore, the present study highlights the valorisation of this side stream as a source of natural phenolic antioxidants by using a green extraction technology as part of a sustainable food system.

## Figures and Tables

**Figure 1 foods-09-00604-f001:**
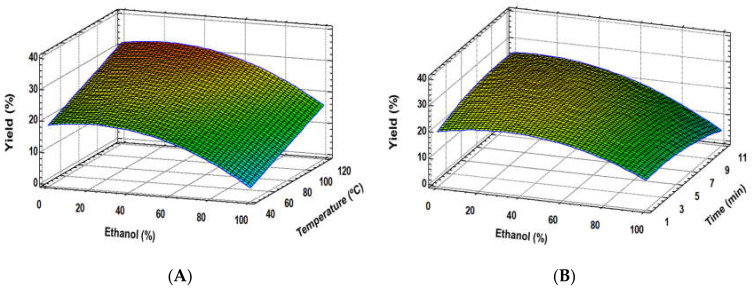
Response surface plots of the extraction yield as affected by independent factors; ethanol (%) *vs*. temperature (**A**), ethanol (%) *vs*. time (**B**).

**Figure 2 foods-09-00604-f002:**
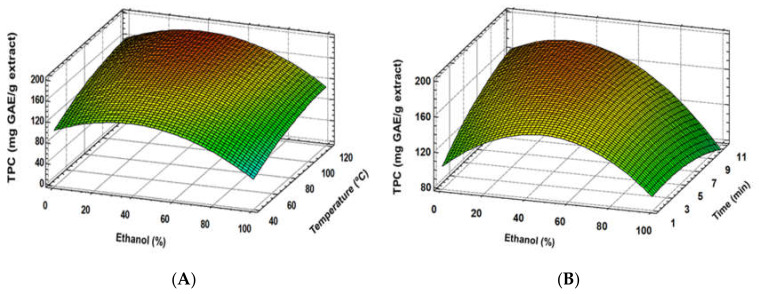
Response surface plots of the TPC as affected by independent factor; ethanol (%) *vs*. temperature (**A**), ethanol (%) *vs*. time (**B**).

**Figure 3 foods-09-00604-f003:**
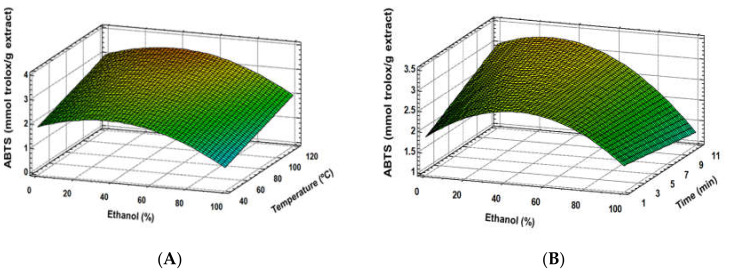
Response surface plots of the ABTS as affected by independent factors; ethanol (%) *vs*. temperature (**A**), ethanol (%) *vs*. time (**B**).

**Figure 4 foods-09-00604-f004:**
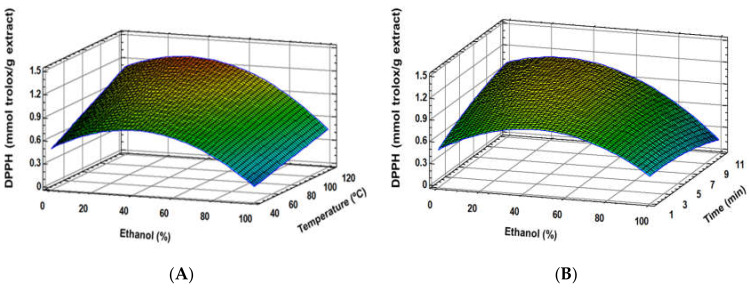
Response surface plots of the DPPH as affected by independent factors; ethanol (%) *vs*. temperature (**A**), ethanol (%) *vs*. time (**B**).

**Figure 5 foods-09-00604-f005:**
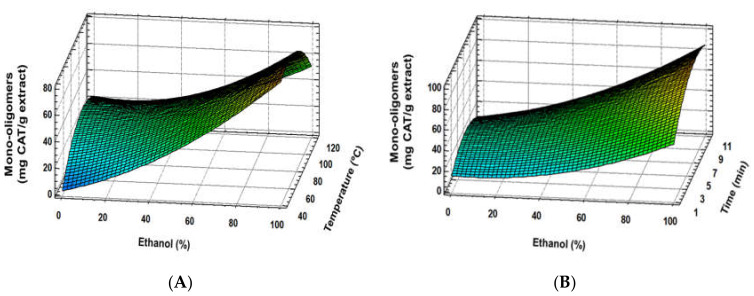
Response surface plots of flavan-3-ol monomer and oligomer content as affected by independent factors; ethanol (%) *vs*. temperature (**A**), ethanol (%) *vs*. time (**B**).

**Figure 6 foods-09-00604-f006:**
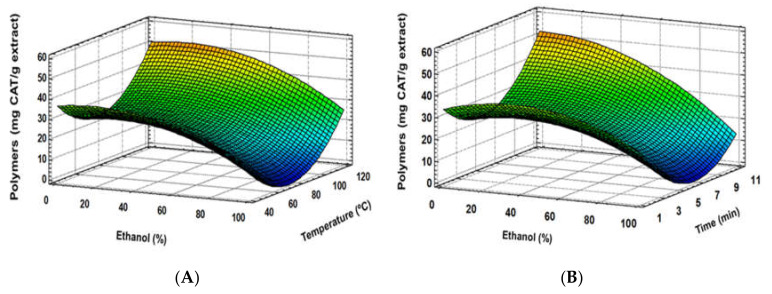
Response surface plots of polymer procyanidins content as affected by independent factors; ethanol (%) *vs*. temperature (**A**), ethanol (%) *vs*. time (**B**).

**Figure 7 foods-09-00604-f007:**
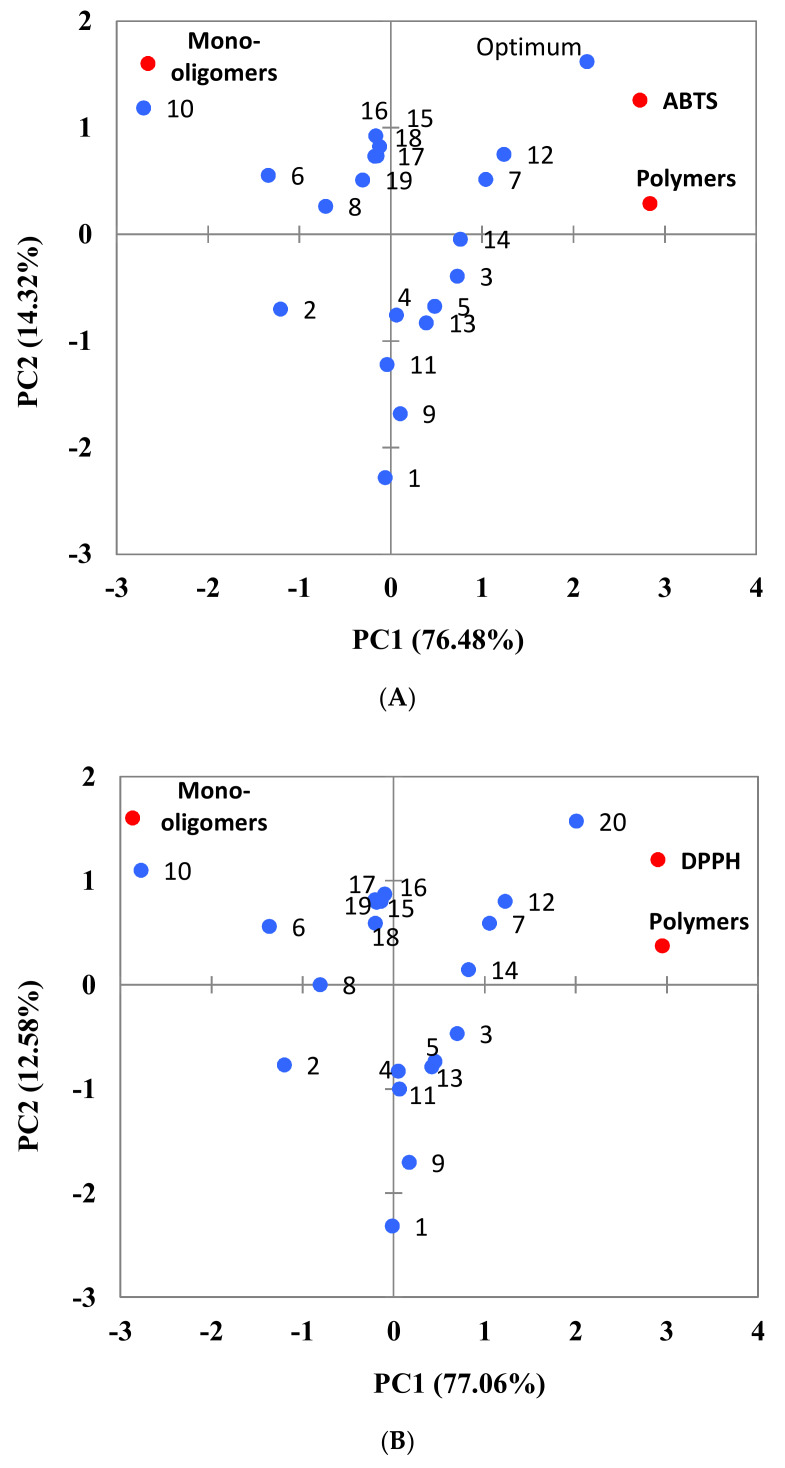
Principal components analysis (PCA) of response variables. Projections of the variables and samples (Biplot) for ABTS (**A**) and DPPH values (**B**).

**Table 1 foods-09-00604-t001:** Coded levels and experimental values of the factors used in the central composite rotatable design.

Factor	Coded Symbol	Coded Levels				
		**−1.68**	**−1**	**0**	**1**	**1.68**
Ethanol concentration (%)	Et	0	20	50	80	100
Temperature (°C)	T	40	56	80	104	120
Time (min)	t	1	3	6	9	11

**Table 2 foods-09-00604-t002:** Experimental design and experimental response variable data.

Factor				Response Variables						
Run	X_1_, Ethanol	X_2_, Temperature	X_3_, Time	Yield		Total Phenolic Compounds (TPC)	ABTS	DPPH	Total Flavan-3-ol Mono- and Oligomers	Total Polymer Procyanidins
	(%)	(°C)	(min)	(g Extract/100 g Stem)		(mg GAE/g Extract)	(mmol Trolox/g Extract)	(mmol Trolox/g Extract)	(mg catechin/g Extract)	(mg catechin/g Extract)
**1**	20	56	3	21.5	118.1		2.06	0.65	19.53	30.02
**2**	80	56	3	9.4	114.0		1.89	0.55	52.72	14.55
**3**	20	104	3	27.5	164.0		2.89	0.97	24.48	42.64
**4**	80	104	3	23.3	145.9		2.50	0.81	31.51	32.10
**5**	20	56	9	21.0	148.7		2.63	0.86	27.16	41.97
**6**	80	56	9	10.3	113.2		1.94	0.56	69.97	21.26
**7**	20	104	9	28.5	178.7		3.33	1.19	26.59	44.58
**8**	80	104	9	14.8	135.2		2.42	0.72	52.64	19.65
**9**	0	80	6	24.9	147.8		2.54	0.87	17.81	21.68
**10**	100	80	6	12.4	87.4		1.39	0.29	98.73	4.06
**11**	50	40	6	18.8	129.4		2.24	0.77	30.67	34.53
**12**	50	120	6	30.6	186.6		3.43	1.22	26.51	49.78
**13**	50	80	1	21.2	147.8		2.60	0.88	26.26	38.42
**14**	50	80	11	24.8	172.7		2.96	1.06	27.73	44.51
**15**	50	80	6	22.8	167.6		2.98	1.00	47.45	22.05
**16**	50	80	6	25.0	171.8		3.00	1.05	45.43	21.62
**17**	50	80	6	24.3	167.9		2.94	1.03	45.67	21.30
**18**	50	80	6	24.1	169.7		2.95	0.98	45.34	21.82
**19**	50	80	6	24.0	160.2		2.77	1.01	46.65	21.48

**Table 3 foods-09-00604-t003:** Polynomial equations and statistical parameters of the fitted models obtained for response variables.

Variable	Polynomial Equation of Fitted Model	*R* ^2^	Lack-of-Fit (p-Value)
**Yields** (g extract/g stem)	Y = −2.05097 + 0.188423(Et) + 0.24681(T) + 2.91554(t) − 0.00271887(Et)^2^ − 0.0115812(Et × t) − 0.0144409(T × t) − 0.0984341(t)^2^	0.951	0.15
**TPC** (mg GAE/g extract)	Y = −15.954 + 2.10974(Et) + 1.94514(T) + 10.808(t) − 0.0212643(Et)^2^ − 0.0807058(Et × t) − 0.00779276(T)^2^ − 0.408779(t)^2^	0.97	0.21
**ABTS** (mmol Trolox/g extract)	Y = 0.578714 + 0.0369343(Et) + 0.0142628(T) + 0.112674(t) − 0.000382393(Et)^2^ − 0.00147263(Et × t)	0.97	0.33
**DPPH** (mmol Trolox/g extract)	Y = −0.238343 + 0.0209687(Et) + 0.00766806(T) + 0.0851506(t) − 0.000183361(Et)^2^ − 0.0000407129(Et × T) − 0.00071712(Et × t) − 0.00273786(t)^2^	0.98	0.19

**Table 4 foods-09-00604-t004:** Optimal extraction conditions, experimental and estimated values for response variables in the optimum extract.

	Optimal Conditions			Optimal Extract Values (30% Et, 120 °C, 10 min)	
	Et (%)	T (°C)	T (min)	Experimental	Estimated
**Yields** (g extract/100 g stem)	25	120	4.5	28.9	29.2
**TPC** (mg GAE/g extract)	30	120	10	187.3	192.4
**ABTS** (mmol Trolox/g extract)	27	120	11	3.69	3.81
**DPPH** (mmol Trolox/g extract)	22	120	11	1.32	1.37

**Table 5 foods-09-00604-t005:** HPLC-PAD-ESI-MS phenolic analysis of the optimal grape stem extracts (mg compound/g dry extract).

Phenolic Compound	UV-Vis Max.	[M − H]^−1^	[M + H]^+1^	MS/MS Fragments	mg/g dry Extract
No Flavonoids					
Hydroxybenzoic acids					
Gallic acid	270	169		125	0.541 ± 0.029
Protocatechuic acid	260/290	153		117	0.008 ± 0.000
Monogalloyl glucoside	257/298	331		169	<LOQ
4-Hydroxybenzoic acid	256	137			0.048 ± 0.001
Vanillic acid	259/292	167		153	0.224 ± 0.010
Syringic acid	278	197		183	0.202 ± 0.015
Ethyl gallate	277	197		169	0.010 ± 0.001
Ellagic acid	256/353	301		229	0.073 ± 0.004
Hydroxycinnamic acids					
*trans*-caftaric acid	296/328	311		179	0.357 ± 0.003
*trans*-caffeic acid	300/324	179		161	0.006 ± 0.000
4-Coumaric acid	290/310	163		119	0.004 ± 0.000
3-Coumaric acid	289/309	163		119	0.003 ± 0.000
Coumaroyl-*O*-glucoside	280/308	325		163,119	0.003 ± 0.000
*trans*- ferulic acid	298/322	193		149	0.008 ± 0.000
Stilbenes					
*trans*-Piceid	295/324	389		227	0.016 ± 0.000
*trans*-Resveratrol	303/328	227		185	0.141 ± 0.003
*ε*-viniferin	262/308/322	453		359	0.879 ± 0.065
*cis*-resveratrol trimer	286	679		585	0.031 ± 0.002
*trans*-resveratrol trimer	296/320	679		587,575	0.012 ± 0.001
*trans*-resveratrol trimer	288/326	679		587,575	0.042 ± 0.003
*trans*-resveratrol tetramer	306/316	905		811	0.136 ± 0.011
*trans*-resveratrol tetramer	306/316	905		811	0.086 ± 0.007
*cis*-resveratrol tetramer	284	905		811,717	0.038 ± 0.003
*trans*-resveratrol tetramer	306/318	905		811,799	<LOQ
Flavonoids					
Flavan-3-ols					
Catechin	278	289		245	2.422 ± 0.034
Epicatechin	278	289		245	1.293 ± 0.039
Epicatechin gallate	280	441		289,169	0.245 ± 0.005
Procyanidin B1	278	577		425	1.410 ± 0.034
Procyanidin B2	278	577		425	0.015 ± 0.003
Procyanidin B3	278	577		425	0.349 ± 0.025
Procyanidin B4	279	577		425	0.036 ± 0.002
Procyanidin B7	280	577		425	0.025 ± 0.003
Procyanidin C1	280	865		577	0.016 ± 0.001
Flavonols					
Kaempferol-3-*O*-glucoside	287/358	447		285	<LOQ
Quercetin-3-*O*-galactoside	256/354	463		301	0.047 ± 0.000
Quercetin-3-*O*-rutinoside	256/354	609		301	0.029 ± 0.000
Quercetin-3-*O*-glucuronide	252/354	477		301	1.425 ± 0.001
Quercetin-3-*O*-glucoside	254/354	463		301	0.106 ± 0.004
Quercetin	256/368	301			0.005 ± 0.000
Anthocyanins					
Delphinidin-3-*O*-glucoside	292/535		465	303	<LOQ
Cyanidin-3-*O*-glucoside	290/530		449	287	0.010 ± 0.001
Malvidin-3-*O*-glucoside	293/537		493	331	0.079 ± 0.002

LOQ: limit of quantification

**Table 6 foods-09-00604-t006:** Characteristics and structural composition (percent in moles) of procyanidin fraction from optimal stem PLE extract.

Terminal Units (%)			Extension Units (%)				mDP	Galloilated Units (%)
Cat	EC	ECG	Cat	EC	ECG	EGC		
6.32	1.07	0.81	8.27	71.27	11.59	0.68	12.22	12.40

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
