# Peer review of "Valorisation of Grape Stems as a Source of Phenolic Antioxidants by Using a Sustainable Extraction Methodology"

_foods, 2020, doi:10.3390/foods9050604_

Round 1
Reviewer 1 Report
I don't have any comments for this manuscript.
The study was well planned and decribe. It could be a little
incomprehensible due to many factors that were taken into account,
but if we just focus on the description it is clear. The authors have properly described the results obtained, especially statistical analysis is worth praising.
Author Response
Response to Reviewer 1 comments
English language and style have been improved according reviewer comments (see Line 17, Line 34, Line 50, etc.)

Reviewer 2 Report
The manuscript ‘Valorisation of grape stems as a source of phenolic antioxidants by using a sustainable extraction methodology’ proposes a study to determine the most efficient extraction procedure (solvent, time and temperature), based on pressurised liquid extraction and ethanol:water mixture as green solvent, to extract the antioxidant compounds from grape steams. Moreover, the phenolic composition was studied to evaluate the relationship between these compounds and the antioxidant activity.
The research is original and fits with the scope of the journal, it also enhances the use of waste products for the extraction of antioxidant compounds. The manuscript is clear and the study is well articulated and described.
Author Response
Response to Reviewer 2 comments
English language and style have been improved according reviewer comments (see Line 17, Line 34, Line 50, etc.)

Reviewer 3 Report
Comments:
- Author mentioned that they have identified “Forty-two phenolic compounds were identified in the optimal extract, mainly polymer procyanidins and, to a lesser extent, monomers and oligomers of flavan-3-ols, quercetin-3-O-glucuronide, viniferin, gallic and caftaric acid. It would be interesting if author can provide LC-MS chromatograpgy for the same.
- The author has to explain why they used “Pressurized liquid extraction with ethanol: water mixtures” and why they didn’t not compare it with other solvent with other extraction method.
- Why author have chosen to determine the phenolic compounds by NP-HPLC and what are those phenolic compounds?
Author Response
Response to reviewer 3 comments
English language and style have been improved according reviewer comments (see Line 17, Line 34, Line 50, etc.)
Point 1. Author mentioned that they have identified “Forty-two phenolic compounds were identified in the optimal extract, mainly polymer procyanidins and, to a lesser extent, monomers and oligomers of flavan-3-ols, quercetin-3-O-glucuronide, viniferin, gallic and caftaric acid. It would be interesting if author can provide LC-MS chromatography for the same.
Response:
A figure with HPLC-PAD is attached as information to the reviewer. We can include this chromatogram in the manuscript, but this means that we have to re-order the analyzed compounds from Table 5 in order of elution and this will change the strategy of exposition of the results we would like to maintain, as we have quantitative results and we think that is mor important to show the analyzed compounds by groups and define their whole amounts. In the other way this will be not possible. Moreover, it should be noted that the article already includes 6 tables and 6 figures, including Table 5 that shows the phenolic analysis.
Point 2. The author has to explain why they used “Pressurized liquid extraction with ethanol: water mixtures” and why they didn’t not compare it with other solvent with other extraction method.
Response: Although some organic solvents have been proposed for phenolic extraction from grape products, aqueous methanol has been widely considered as the most efficient solvent for that purpose. However, as a toxic organic solvent, nowadays, the use of methanol should be replaced by other environmental friendly solvents. In this sense, ethanol has been proposed to replace methanol, because of the closeness of the physical properties of both solvents. Changes in lines 228-230 have been implemented in order to improve this comment. Therefore, we know for previous studies with other grape products, that aqueous methanol is the most appropriate solvent of phenolic extraction, more than acetone, of even more than ethanol (although it depends on the type of phenolic compounds). However, the goal of the article is focus on replacement of the methanol for other sustainable solvent, and ethanol it is proposed as the most suitable.
Moreover, many articles have shown that PLE or UAE are more efficient extraction techniques than SLE. In previous studies carried out in our laboratory, we already saw that PLE was a more efficient extraction technology compared to conventional SLE for other grape stems. Therefore, we preferred to focus the study directly on the development of optimal extraction conditions by using aqueous methanol mixtures by using PLE.
Point 3. Why author have chosen to determine the phenolic compounds by NP-HPLC and what are those phenolic compounds?
NP-HPLC is used here because it allows to identify and quantify oligomeric procyanidins in an easier way. In fact, this method is used only for determination of total monomer, total oligomer and moreover, total polymer flavanols that elute as a singular peak at the end of the chromatogram. You can read a previous paper of our research group (Muñoz-Labrador et al. 2019) where a very similar chromatogram is available, together with some explanation about how this chromatogram should be interpreted. The rest of phenolics are determined by RP-HPLC. To make clearer this statement, we have done some modifications in the corresponding paragraph 2.6 (Material & Methods section), that are matched in red. However, we can attach a chromatogram to illustrate this if you like.

Reviewer 4 Report
Paper: Valorisation of grape stems as a source of phenolic antioxidants by using a sustainable extraction methodology described by Nieto et al. is typically paper about extraction methods and results: extraction and analysis compounds after this process. Authors used the Response surface methodology for optimal extraction conditions, it's not new concept. In literature, this method is analyzed very often. Therefore the novelty of this paper is only to use grape stems as a source of phenolic antioxidant compounds.
Additionally comments:
- Delete “.” After title
- present all results of antioxidant activity as mmol Trolox/g
- - -O- in quercetin-3-O-rutinoside and other compounds should be -O-
- Results present in Table 1 and 3 should be better discussed and compared with literature because sentences like: Similar results were found in other studies, where ethanol: water mixtures behavior was analyzed […]. – is not merit but only for information
- Additionally for correlation Authors should be present these parts as PCA analysis
- How Authors identification these compounds, how Authors knows that is -3-O- in Coumaroyl-3-O-glucoside or similar to cis- or trans- position after only analysis by LC-MS
- For Hyperoside and Rutin – use the full name
- How Authors know that compounds with m/z 577 are i.e. Procyanidin B7 or B4, etc.
- Do the Authors taste this phenolic? and try to make some composition of some products?
Author Response
Response to reviewer 4 comments
Point 1. Valorisation of grape stems as a source of phenolic antioxidants by using a sustainable extraction methodology described by Nieto et al. is typically paper about extraction methods and results: extraction and analysis compounds after this process. Authors used the Response surface methodology for optimal extraction conditions, it's not new concept. In literature, this method is analyzed very often. Therefore the novelty of this paper is only to use grape stems as a source of phenolic antioxidant compounds.
Response: We agree with the reviewer’s comment regarding Response surface methodology. Moreover, in our opinion experimental design and response surface methodology should be mandatory for any article focus on optimizing extraction conditions, although most of the currently articles do not use it.
Furthermore, in our opinion the main novelty of our article is of course the use of grape stems, but also the extraction conditions, with the use of GRAS solvents and by using a more efficient extraction technology, such as PLE. All these conditions allow developing a more sustainable extraction procedure, in addition to a valorisation of grape stem as antioxidant phenolics source. Moreover, an exhaustive analysis of the phenolic antioxidants of this side stream is displayed.
Point 2. Delete “.” After title
Response: Done
Point 3. present all results of antioxidant activity as mmol Trolox/g
Response: Done in line with the comment (e.g. line 19)
Point 4. -O- in quercetin-3-O-rutinoside and other compounds should be -O-
Response: -O- was corrected throughout the manuscript.
Point 5. Results present in Table 1 and 3 should be better discussed and compared with literature because sentences like: Similar results were found in other studies, where ethanol: water mixtures behavior was analyzed […]. – is not merit but only for information
Response: A deeper discussion has been done as can be seen in lines 299-316.
Point 6. Additionally for correlation Authors should be present these parts as PCA analysis
Response: PCA has been included in the manuscript (Figure 7, lines 435-446)
Point 7. How Authors identification these compounds, how Authors knows that is -3-O- in Coumaroyl-3-O-glucoside or similar to cis- or trans- position after only analysis by LC-MS
Response: We agree with the Reviewer’s comment. Coumaroyl-3-O-glucoside was written erroneously. This compound was identified tentatively, so we don’t know the exact position of the glycosylic linkage. It is rectified in Table 5.
With respect to the cis- or trans- position of the stilbene derivatives, there is no way of mistake, because the UV spectra of the trans- forms are very different from those of the cis-forms. Moreover, we took advantage of this and introduced also the ‘trans-‘ position of the hydroxycinnamic acids, identified and quantified in our manuscript. In this case they all are trans- forms and are identified on base of commercial reference substances
Point 8. For Hyperoside and Rutin – use the full name
Response: In line with the comment quercetin-3-O-galactoside and querceti-3-O-rutinoside have been used. See Table 5 and line 437.
Point 9. How Authors know that compounds with m/z 577 are i.e. Procyanidin B7 or B4, etc.
Response: We have these dimers purified in a previous work carried out by counter current chromatography in collaboration with the group of Prof. Peter Winterhaleter from the Technical University of Braunshweig. This is specified clearer in the Material and Methods paragraph by citation of a previous work where this purification is explained [40].
Point 10. Do the Authors taste this phenolic? and try to make some composition of some products?
Response: A preliminary short test was conducted to know the possible acceptance of the optimal extract and its further possible uses as an ingredient but have not done classic tasting with trained panellist in order to published it. The participants indicated that the sample powder was characterised by a strong astringent feeling and a soft bitter taste, according to the high content of polymer procyanidins. Increasing concentrations of the sample diluted in ultrapure water were tested also. The participants indicated that concentrations over 8 mg/mL were absolutely accepted, reaching, at this concentration, the first astringent feeling in mouth. By increasing the sample concentration, it could be determined that the maximum acceptance concentration of the sample by the participants was at 20 mg/mL. A complete sample solubilisation was observed at this concentration. Therefore, it was concluded that the sample could be a potential ingredient for functional beverages or nutraceuticals, where higher sample concentrations may be used since the beverage flavour could reduce the astringent impact of the sample.

Round 2
Reviewer 4 Report
Revised version after correction is suitable for publishing.